# Socioeconomic determinants of virtual care use among people living with HIV in a clinical cohort in Ontario, Canada: A cross-sectional study

Nadia Rehman [1*], Lawrence Mbuagbaw [1,2,3,4,5,6], Dominik Mertz [1,7,8☯], Giulia M. Muraca [1,9☯], Aaron Jones[1], on behalf of the Ontario HIV Treatment Network Cohort Study[¶]

1 Department of Health Research Methods, Evidence, and Impact, McMaster University, Hamilton, Ontario, Canada, 2 Department of Anesthesia, McMaster University, Hamilton, Ontario, Canada, 3 Centre for Development of Best Practices in Health (CDBPH), Yaoundé Central Hospital, Yaoundé, Cameroon, 4 Department of Pediatrics, McMaster University, Hamilton, Ontario, Canada, 5 Biostatistics Unit, Father Sean O'Sullivan Research. Centre, St Joseph's Healthcare, Hamilton, Ontario, Canada, 6 Division of Epidemiology and Biostatistics, Department of Global Health, Stellenbosch University, Cape Town, South Africa, 7 Faculty of Health Sciences, McMaster University, Hamilton, Ontario, Canada, 8 Department of Medicine, McMaster University, Hamilton, Ontario, Canada, 9 Department of Obstetrics and Gynecology, McMaster University, Hamilton, Ontario, Canada

¶ Membership of the Ontario HIV Treatment Network (OHTN) is provided in the Acknowledgments.
☯ These authors contributed equally to this work.
* rehmann@mcmaster.ca

## Abstract

### Background

Retention in HIV care is essential for achieving optimal health outcomes and extending life expectancy among people living with HIV. However, socioeconomic challenges frequently hinder consistent engagement in care. Virtual care offers a potential solution by enhancing timely access to HIV services and addressing these barriers.

### Objectives

We aimed to examine the association between socioeconomic factors and the use of virtual care among people living with HIV (PLHIV) in a clinical cohort in Ontario, Canada.

### Methods

We analyzed 2022 data from the Ontario HIV Treatment Network Cohort Study (OCS), coinciding with the initial rollout of virtual care. The OCS is a multi-site cohort comprising patients from 15 HIV clinics, with data obtained from medical charts, interviews, and linkage to provincial public health lab (PHOL) records for viral load testing. We employed multinomial logistic regression to identify predictors of care mode: virtual, in-person, or a hybrid (virtual and in-person).

**Data availability statement:** The data used in this study were obtained from the Ontario HIV Treatment Network Cohort Study (OCS). Due to privacy and confidentiality agreements, the authors cannot share the dataset or raw data publicly. However, data access may be requested directly from the OHTN by submitting a Research Application Process (RAP) form. Requests can be made by emailing ocs@ohtn. on.ca.

**Funding:** The author(s) received no specific funding for this work.

**Competing interests:** The authors have declared that no competing interests exist.

## Results

The study included 1,930 participants. Of these, 19% (n = 367) received virtual care, 45.6% (n = 900) received in-person care, and 34.3% (n = 663) accessed hybrid care services. The median participant age was 55 years [Q1; Q3: 45; 62], and 78% (n = 1,131) identified as men who have sex with men (MSM). Compared to Toronto residents, individuals living in Southwestern Ontario had higher odds of using virtual care (adjusted OR (AOR) 1.67, 95% CI: 1.13, 2.47). Conversely, females (AOR = 0.59; 95% CI: 0.40, 0.88), heterosexual men (AOR = 0.64; 95% CI: 0.45, 0.92), residents of Eastern Ontario (AOR = 0.42; 95% CI: 0.26, 0.68), individuals with a high school education (AOR = 0.67; 95% CI: 0.46, 0.98), those with an annual gross income of CAD $71,000–90,000 (AOR = 0.59; 95% CI: 0.38, 0.91), and individuals diagnosed with HIV within the last 10 years (AOR = 0.59; 95% CI: 0.39, 0.91) were less likely to use virtual care. Participants experiencing any level of depression were more likely to use hybrid care services.

## Conclusion

Virtual care was introduced during the COVID-19 pandemic to enhance healthcare access in Ontario. Its adoption varied based on socioeconomic and health-related factors in the OCS cohort. Ongoing research is needed to assess these patterns beyond the pandemic context..

## Introduction

Retention in HIV care is essential for the effective management of people living with HIV (PLHIV). While antiretroviral therapy (ART) has significantly reduced HIV-related morbidity and mortality, inconsistent retention in care can lead to low adherence to ART, drug resistance, poor health outcomes, and a greater risk of HIV transmission. Ensuring continuous care is vital for optimizing treatment outcomes [1–3].

The HIV burden is disproportionately prevalent among specific key populations and racial groups [4,5]. In Ontario, 2022 data show that 56.8% of new HIV cases occurred among gay, bisexual, and men who have sex with men (MSM), 29.8% among individuals of African, Caribbean, and Black descent, and 10.6% among people with a history of intravenous drug use [6]. These populations often face various socioeconomic disparities and individual challenges, including substance use, mental health conditions, stigma, and discrimination, which contribute to suboptimal retention in healthcare services and adherence to ART [7–9]. Furthermore, structural barriers within the patient-provider relationship and clinic settings can disrupt the continuity of HIV care [10,11]. In Ontario, 2020 data show that retention in HIV care varied across regions, with overall 89.2% of people living with diagnosed HIV retained in care, and even lower retention in some areas such as Northwestern Ontario (67.6%) [6].

The COVID-19 pandemic accelerated the adoption of virtual care across Ontario's healthcare systems [12]. Due to its flexibility, virtual care has the potential to improve

access to HIV services by addressing socioeconomic barriers. Although concerns persist, such as connectivity issues and the inability to conduct objective tests, participants reported benefits including increased privacy, reduced transportation needs, and greater scheduling flexibility [13,14].

Our recent analysis of 2022 data from the Ontario HIV Treatment Network Cohort Study (OCS) found that virtual care was associated with improved adherence to ART and better viral suppression. However, the cross-sectional design limited our ability to infer causality or assess differences in impact across sociodemographic groups [15]. A retrospective study from the Johns Hopkins HIV Clinical Cohort (2010) compared clinic visit uptake over one year of in-person care with one year of exclusively virtual care during the COVID-19 pandemic. Although overall visit uptake declined under the virtual care model, certain groups, such as older adults, men, Black patients, and individuals with a history of substance use disorder, demonstrated improved retention. However, the findings of this study are limited by the confounding influence of government mandates on healthcare utilization during the pandemic period [16].

Given the relative novelty of virtual care, it is important to understand the factors influencing the choice of care modality, as this insight can help address barriers to care retention. Given this gap, our study aimed to examine whether various sociodemographic, structural and health-related factors were associated with the use of virtual care among PLHIV in the 2022 OCS cohort, when COVID-19 restrictions had been lifted, and virtual care had been formally adopted as a standard care option in Ontario, Canada [17]. Recognizing the established link between patient satisfaction, quality of life (QoL), and clinical outcomes, this study further aimed to evaluate patient satisfaction and explore barriers to HIV care, as captured through the OCS questionnaire [18]. To ensure the study is grounded in the experiences of those it aims to serve, we partnered with Realize, a Canadian charity supporting people with HIV, to establish a community advisory board (CAB) [19]. By incorporating the voices and insights of PLHIV, we aim for findings that resonate with communities [20,21].

## Methods

### Data sources study design

We conducted a cross-sectional study using data collected in 2022 from participants enrolled in the OCS [22]. The OCS is an ongoing, multisite clinical cohort of people living with HIV who are receiving HIV care in Ontario, Canada. Detailed descriptions of the cohort have been published previously [23]. Briefly, the source population of the OCS includes individuals aged 16 years and older living with HIV who receive medical care within Ontario's publicly funded healthcare system. Participants are recruited from 15 HIV care sites across the province, including hospital-based outpatient clinics and community-based practices. The OCS integrates longitudinal clinical data with patient-reported information using standardized data collection procedures.

Clinical data are obtained through manual and electronic abstraction of medical records from routine follow-up visits and are supplemented by linkage with Public Health Ontario Laboratories, the sole provincial provider of HIV-related laboratory testing, including viral load measurements. These clinical data are complemented by standardized questionnaires administered at enrollment and annually thereafter, which collect detailed sociodemographic, socioeconomic, psychosocial, and behavioural information, including employment, income, mental health, substance use, and healthcare utilization [24].

For the present analysis, we used cross-sectional data from the 2022 OCS questionnaire, which included detailed measures of HIV care delivery modalities, including in-person and virtual care via telephone or video, as well as relevant demographic, socioeconomic, and clinical covariates.

**Ethics.** All participants enrolled in the OCS cohort provided written informed consent, and the study protocol and consent forms were approved by the Research Ethics Boards (REBs) of the University of Toronto and each participating site. The de-identified data were first accessed on November 3, 2023, following approval from the OHTN Cohort Study Governance Committee [20,25].

## Study population

We used cohort data available as of 2022 to define our study population. Participants were eligible for this analysis if they were 16 years or older, had at least one appointment with their HIV physician in 2022, and had completed the OCS questionnaire. Individuals with incomplete information regarding the type of care received were excluded from the analysis.

## Measures/outcomes

HIV care was classified into three mutually exclusive categories: (i) in-person clinic visits; (ii) virtual care via telephone or video call; and (iii) a combination of virtual and in-person services (hybrid). Virtual care was defined as consultations with an HIV care physician by telephone or video call, as defined in the 2022 OCS Questionnaire.

Baseline demographic and clinical covariate data were extracted from the 2022 OCS questionnaire to examine potential predictors of retention in 2022. Age was analyzed both as a continuous and categorical variable, divided into four groups: ≤ 30, 31–40, 41–50, and > 50 years. Sex and sexual orientation were combined into a single variable with three categories: females, MSM, and heterosexual men [7]. Other covariates are defined as follows: race or ethnicity (African, Caribbean, or Black versus White versus others); language fluency, derived by combining two variables: Canadian-born and immigrants living ≥ 10 years, defined as participants with proficient English-speaking skills, versus immigrants living for < 10 years, defined as having below-average English-speaking skills; relationship status, categorized as stable (married, living common-law, or in a committed relationship) versus unstable (widowed, separated/divorced, or single); regions in Ontario categorized into four areas (Greater Toronto Area, Eastern Ontario, Northern Ontario, Southwestern Ontario); education level (elementary versus high school versus college versus higher education); employment status (employed versus unemployed); and annual personal income (≤ $50,000, $51,000-$70,000, $71,000-$100,000, and> $100,000).

Barriers to virtual care were defined as privacy or lack of privacy, depending on the availability of a private space for attending the virtual visit. The housing situation was categorized as stable (condo, housing facility, shelter, or room in a motel or hotel) or unstable (correctional facility, couch surfing, or living on the street).

Clinical covariates extracted include adherence to ART (self-reported), categorized as optimal adherence (≥ 95%: never skipped or skipped more than three months ago) and suboptimal adherence (< 95%, defined as missed doses within the past week, one to two weeks, two to four weeks ago, or one to three months ago) [26]; viral load was defined as suppressed (≤ 40 copies/mL) or unsuppressed (> 40 copies/mL) [27]; alcohol abuse, measured using the Alcohol Use Disorder Identification Test (AUDIT-10), with harmful alcohol use defined as a score of ≥ 8 regardless of gender/sex [28]; depression, measured by the Patient Health Questionnaire (PHQ) scale, with nine items, with depression defined as normal (0–2), mild (3–5), moderate (6–8), and severe depression (9–12) [29]; health-related QoL, assessed using the Short Form 36 Health Survey with two components: Mental Component Summary Score (MCS) and Physical Component Summary Score (PCS) [30]; smoking, classified as heavy smokers (at least 20 cigarettes daily) vs. moderate smokers [31]; diagnosis of mental health comorbidities, based on self-report in the OCS questionnaire; and stigma, assessed using a 10-item HIV-related stigma scale, categorized into four major components: personalized stigma, worries about disclosure of status, negative self-image, and sensitivity to public reactions regarding HIV status. Individuals who responded with "agree" or "strongly agree" were identified as experiencing stigma in at least one of the four components [32].

## Patient satisfaction survey

The OCS questionnaire includes a satisfaction survey on virtual and in-person HIV care, divided into three sections:

**Experience with virtual care.** Participants experience with virtual care was assessed across 10 elements: (1) physician's time; (2) time saved; (3) cost savings; (4) involvement in decision-making; (5) sense of safety; (6) ease of technology use; (7) addressing health concerns; (8) communication of health issues; (9) preference for future virtual care; and (10) opportunity to ask treatment-related questions. Responses were rated on a 5-point Likert scale: strongly agree, agree, neither agree nor disagree, disagree, and strongly disagree, with higher scores indicating greater satisfaction.

**Experience with HIV care provider.** Participants experience with HIV care provided was evaluated across 13 elements, including (1) familiarity with patient history; (2) listening skills; (3) language proficiency; (4) clarity in explanations; (5) sensitivity to needs; (6) dignity in treatment; (7) ability to provide instructions; (8) responsiveness to questions; (9) concern for medication coverage; (10) involvement in treatment; (11) adequate time spent; (12) confidence in the provider; and (13) confidence in provider knowledge. Responses used a 3-point Likert scale: excellent/very good, good, and fair/poor.

**Experience with clinical practice.** Participants experience with the clinical practice was assessed on five elements: (1) ease of accessing the visit; (2) wait time; (3) level of privacy; (4) consistency of communication; (5) quality of staff interactions: and (6) overall experience in accessing the care. Ratings followed the same 3-point Likert scale.

**Barriers to HIV care.** We also evaluated barriers to HIV care identified in the OCS questionnaire, including factors related to technology and access (e.g., internet availability), barriers to access (e.g., travel time to the clinic), healthcare utilization, and experiences of discrimination.

## Data analysis

We calculated descriptive statistics, reporting proportions for categorical variables and medians with interquartile ranges (IQR) for continuous variables. We compared continuous covariates using the Kruskal-Wallis test and categorical variables using chi-square tests. We considered a p-value of < 0.05 to indicate statistical significance.

Prior to fitting the multinomial logistic regression models, diagnostic and preparatory analyses were conducted to assess model stability and specification. Multicollinearity among independent variables was evaluated using variance inflation factors (VIFs) derived from a linear proxy model that included the same set of covariates; all VIFs were below 2, indicating minimal multicollinearity among predictors [33]. Influential observations were assessed using Cook's distance; although a small number of observations exhibited elevated influence values, these reflected plausible participant characteristics and did not materially affect model estimates [34].

Theoretical and clinical considerations guided model specification. Variables such as age, depression severity, and viral load were a priori categorized to capture potentially nonlinear relationships supported by the existing literature [27,29]. The only covariates modelled as continuous variables were the Physical Component Summary (PCS-12) and Mental Component Summary (MCS-12) scores, which are validated continuous measures of health-related quality of life [30,35]. Plausibility and potential departures from linearity of continuous variables were evaluated using graphical methods, including inspection of distributions, residual diagnostics from proxy regression models, and smoothed plots. These assessments did not reveal meaningful departures from linearity or evidence that inclusion of interaction terms or higher-order nonlinear effects would improve model fit. Accordingly, a parsimonious main-effects multinomial logistic regression model was retained.

The multinomial logistic regression model relies on the Independence of Irrelevant Alternatives (IIA) assumption [36]. We assessed the robustness of this assumption using both formal testing and sensitivity analyses. The Hausman–McFadden specification test did not indicate statistically significant violations of the IIA assumption (p > 0.05) [37]. Given the known limitations of this test in finite samples and in the presence of correlated outcome categories, we additionally conducted a sensitivity analysis excluding the hybrid care category and fitted a binary logistic regression comparing virtual versus in-person care [38]. Findings from the sensitivity analysis were consistent with those from the primary multinomial model, indicating that the substantive findings were not materially sensitive to potential violations of the IIA assumption. The outcome categories: in-person care, virtual care, and hybrid care, were defined a priori and reflect distinct modes of healthcare delivery. Covariates with sparse cell counts (housing and stigma) across outcome categories were excluded from the final models, as they resulted in numerical instability during model estimation.

Following model diagnostics and robustness assessments, we fit a three-category multinomial logistic regression model, with in-person care as the reference category, to identify independent correlates of care modality use

[39]. Given the limited prior research on predictors of virtual care use, we included all covariates in the model irrespective of statistical significance, thereby creating an explanatory model. The model results are reported as odds ratios (OR) with 95% confidence intervals (CI). A 95% CI excluding the null value was considered statistically significant.

Since the OCS questionnaire had data missing at random, we performed 10 imputations for the model and combined the results using Rubin's rule [40]. However, the dataset comprised two components, and to preserve unbiased responses and accurately represent participants' preferences, we did not impute the participant survey data [41]. The survey responses and the assessment of barriers to care were summarized using numerical and percentage data. Statistical analysis was performed using R software version 4.4.1.

**Sample size calculations.** As of December 31, 2022, 2155 individuals completed the OCS questionnaire across the 15 OCS sites. The primary outcome is adherence to ART; 692 participants demonstrated suboptimal adherence, and 1293 demonstrated optimal adherence. We planned a study including 1,930 participants after excluding individuals with missing information on the type of care modality used. The sample size resulted in 80% power to detect a difference of 20% or greater between participants with suboptimal and optimal [42] The Type I error probability associated with the test of the null hypothesis is 0.05 for a two-tailed chi-squared statistic. (PS: Power and Sample Size Calculation version 3.1.2, 2014 by W.D. Dupont & W.D. Plummer Jr).

## Results

In 2022, a total of 2,155 people completed the OCS questionnaire, with 1,930 participants reporting the type of care they received. Of these, 19% (n = 367) of HIV care visits were conducted virtually, 46% (n = 900) were in-person visits, and 34% (n = 663) used hybrid services. The median age of the participants was 55 years (IQR: 45–62). A notable 78% (n = 1,497) of participants were men, with the majority identifying as MSM, making up 51.7% (1,119/1,930) of the sample. Among the MSM participants, 63% (708/1,119) optimally adhered to ART, with a median age of 57 years (IQR: 47–63).

In terms of racial demographics, participants identifying as White represented the largest group at 61% (1,172/1,930). The usage of the three different care modalities varied across participant characteristics.

Significant differences were observed in the proportions of in-person, virtual, and hybrid care use by sex, race, region, language fluency, relationship, and viral load (Table 1).

### Socio-demographic factors associated with the type of care

**Sex.** Compared to MSM, female and heterosexual men participants were less likely to utilize virtual care than in-person care (females: Adjusted odds ratio (AOR) 0.59, 95% confidence interval (CI): 0.40, 0.88; and heterosexual men: AOR 0.64, 95% CI: 0.45, 0.92).

**Region.** Compared to participants residing in Toronto, those in the Eastern region were less likely to receive virtual care (AOR 0.42, 95% CI: 0.26, 0.68) or hybrid care (AOR 0.23, 95% CI: 0.14, 0.37). In contrast, participants from the southwestern region of Ontario had higher odds of attending virtual care (AOR 1.67, 95% CI: 1.13, 2.47) or hybrid visits (AOR 1.67, 95% CI: 1.13, 2.47) compared to in-person care.

**Socio-demographic factors.** Compared to individuals with a university degree, those with a high school education had lower odds of attending virtual visits (AOR 0.67, 95% CI: 0.46, 0.98). Participants with an annual gross income between $70,000 and $100,000 were less likely to attend virtual visits than those earning over $100,000 (AOR 0.59, 95% CI: 0.38, 0.91).

Participants diagnosed with HIV for less than 10 years were more likely to attend virtual visits compared to those diagnosed for more than 10 years. Additionally, participants with mild (AOR 2.20, 95% CI: 1.54, 3.13), moderate (AOR 2.06, 95% CI: 1.28, 3.31), moderate severe (AOR 2.80, 95% CI: 1.49, 5.26), and severe depression (AOR 2.46, 95% CI: 1.13, 5.32) preferred hybrid care services (Table 2).

**Table 1. Comparison of baseline characteristics between participants who accessed HIV care through virtual, in-person or both virtual and in-person care in Ontario, Canada, in 2022.**

| Characteristics | In-person care n (%) | Virtual care n (%) | Hybrid n (%) | Total N (%) | p-value |
|---|---|---|---|---|---|
| No. of participants | 900 (46.6) | 367 (19.0) | 663 (34.4) | 1930 (100) | |
| Median age [Q1; Q3][1] | 55 [44; 63] | 56 [46; 62] | 55 [45; 62] | 55 [45; 62] | 0.772 |
| *Sex* | | | | | **< 0.001** |
| Female | 233 (25.9) | 60 (16.3) | 132 (20.2) | 425 (22.0) | |
| MSM[2] | 459 (50.8) | 253 (70.0) | 419 (63.4) | 1131 (58.6) | |
| Heterosexual male | 203 (22.6) | 49 (13.4) | 110 (16.2) | 362 (18.8) | |
| Missing | 5 (0.55) | 5 (1) | 2 (0.3) | 12 (0.6) | |
| *Race* | | | | | **< 0.001** |
| White | 478 (53.1) | 252 (68.7) | 442 (66.6) | 1172 (60.7) | |
| ACB[3] | 263 (29.3) | 55 (14.9) | 130 (19.6) | 448 (23.2) | |
| Other | 159 (17.6) | 60 (16.3) | 91 (13.7) | 310 (16.1) | |
| *Region* | | | | | **< 0.001** |
| Eastern Ontario | 127 (14.1) | 24 (6.5) | 22 (3.3) | 173 (9.0) | |
| Northern Ontario | 18 (2) | 5 (1.4) | 2 (0.3) | 25 (1.3) | |
| Southwestern Ontario | 99 (11) | 62 (16.9) | 174 (26.2) | 335 (17.4) | |
| Toronto | 656 (72.8) | 276 (75.2) | 465 (70.1) | 1397 (72.4) | |
| *Language fluency* | | | | | **< 0.001** |
| Yes[a] | 282 (31.3) | 98 (26.7) | 176 (26.5) | 556 (28.8) | |
| No[b] | 89 (9.8) | 12 (3.2) | 12 (1.8) | 113 (5.9) | |
| Canadian born | 426 (47.3) | 244 (66.5) | 390 (58.8) | 1060 (54.9) | |
| Neither | 112 (12.4) | 18 (4.9) | 71 (13.4) | 201 (10.4) | |
| *Education* | | | | | 0.246 |
| Elementary | 25 (2.7) | 10 (2.7) | 14 (2.1) | 49 (2.5) | |
| High school | 239 (26.5) | 126 (34.3) | 224 (33.7) | 589 (30.5) | |
| College | 210 (23.3) | 64 (17.4) | 139 (20.9) | 413 (21.4) | |
| Higher education | 363 (40.3) | 162 (44.1) | 284 (42.8) | 809 (41.9) | |
| Missing | 63 (7) | 5 (1.4) | 2 (0.3) | 70 (3.6) | |
| *Employment status* | | | | | 0.285 |
| Employed | 433 (48.1) | 182 (49.6) | 327 (49.3) | 942 (48.8) | |
| Unemployed | 463 (51.4) | 182 (49.6) | 336 (50.7) | 981 (50.8) | |
| Missing | 4 (0.4) | 3 (0.8) | 0 (0) | 7 (0.4) | |
| *Gross annual income (CAD)[4]* | | | | | 0.287 |
| ≤50,000 | 374 (41.5) | 144 (39.2) | 280 (42.2) | 798 (41.3) | |
| 51,000-70,000 | 103(11.4) | 45 (12.2) | 75 (11.3) | 233 (11.6) | |
| 71,000-100,000 | 121 (13.4) | 41 (11.1) | 76 (11.4) | 238 (12.3) | |
| >100,000 | 173 (19.2) | 94 (25.6) | 150 (22.6) | 417 (21.6) | |
| Missing | 129 (14.3) | 43 (11.7) | 82 (12.3) | 254 (13.2) | |
| *Relationship* | | | | | **0.014** |
| Stable | 376 (41.6) | 154 (42.0) | 246 (37.1) | 776 (40.2) | |
| Unstable | 521 (57.8) | 212 (57.8) | 417 (62.9) | 1150 (59.6) | |
| Missing | 3 (0.3) | 1 (0.3) | 0 (0) | 4 (0.2) | |
| *Adherence to ART* | | | | | 0.172 |
| <95% | 295 (32.7) | 137 (37.3) | 247 (37.3) | 679 (35.2) | |
| ≥95% | 600 (66.6) | 225 (61.3) | 412 (62.1) | 1237 (64.1) | |

*(Continued)*

**Table 1.** (Continued)

| Characteristics | In-person care n (%) | Virtual care n (%) | Hybrid n (%) | Total N (%) | p-value |
|---|---|---|---|---|---|
| Missing | 5 (0.5) | 5 (1.3) | 4 (0.6) | 14 (0.7) | |
| *Depression* | 197 (21.8) | 44 (12.0) | 167 (25.1) | 408 (21.1) | 0.190 |
| *Alcohol use disorder syndrome* | | | | | 0.248 |
| No | 67 (7.4) | 34 (9.2) | 65 (9.8) | 166 (8.6) | |
| Yes | 824 (91.6) | 331 (90.1) | 598 (90.1) | 1753 (90.8) | |
| Missing | 5 (0.6) | 1 (0.3) | 5 (0.8) | 11 (0.5) | |
| *Cigarette smoking* | | | | | 0.161 |
| Heavy smoker | 44 (4.88) | 18 (4.90) | 43 (6.48) | 105 (5.4) | |
| Moderate smoker | 168 (18.6) | 56 (15.2) | 115 (17.3) | 339 (17.6) | |
| Neither | 686 (75.7) | 293 (79.8) | 507 (76.1) | 1486 (77) | |
| *Stigma* | | | | | 0.508 |
| Stigmatized | 125 (13.8) | 12 (3.28) | 92 (13.8) | 105 (5.4) | |
| Not stigmatized | 8 (0.88) | 2 (0.54) | 7 (1.05) | 339 (17.6) | |
| Missing | 767 (85.2) | 353 (96.1) | 564 (85.0) | 1486 (77) | |
| *Quality of life; Median [Q1; Q3]* | | | | | |
| MCS[3] | 50.2 [41.2; 57.0] | 49.2 [40.8;55.3] | 50.9 [38.6; 56.3] | 50.2 [49.2; 50.5] | 0.403 |
| PCS[4] | 53.1 [43.8; 57.2] | 53.5 [44.8; 57.3] | 52.7 [43.5; 56.7] | 53.1 [52.7; 53.3] | 0.548 |
| *Viral load* | | | | | **0.020** |
| ≤ 40 | 766 (85.0) | 336 (91.6) | 579 (87.3) | 1681 (87.1) | |
| > 40 | 104 (11.5) | 26 (7.0) | 70 (10.5) | 200 (10.4) | |
| Missing | 30 (0.5) | 5 (1.4) | 14 (2.1) | 49 (2.5) | |

[1]25th Quartile and 75th Quartile.

[2]Men who have sex with men.

[3]African, Caribbean, and Black.

[4]Canadian Dollars.

[5]Mental component summary score.

[6]Physical component summary score.

[a]Immigrant ≥ 10 years.

[b]Immigrant <10 years.

**Patients' satisfaction survey.** The survey assessing patients' attitudes toward their experience with virtual care indicated high satisfaction, with 937 of 1,030 participants (91.0%) responding "agree" or "strongly agree" across all 10 domains of the satisfaction survey (S1 Fig).

The survey on participants' experiences with HIV clinical practices (S2 Fig) and care providers (S3 Fig) indicated high satisfaction, with 90% and 89% of respondents reporting positive experiences, respectively; however, the overall response rate was only 22%.

**Barriers to virtual visits.** The OCS questionnaire identified several barriers to virtual HIV care. Among participants who preferred in-person visits, 69.3% (n = 624/900) preferred face-to-face consultations, and 40% (n = 355/900) reported not being offered virtual care. A lack of private space for virtual visits was reported by 33.7% of participants (652/1,930), including 17.1% (112/652) of those in the virtual care group.

Significant differences were observed in the proportion of participants seeking care from providers other than their usual HIV care provider: 41.2% of hybrid care users (n = 454/630), 39.2% of in-person care users (n = 600/900), and 54.9%

**Table 2. Factors associated with virtual care and hybrid visits in HIV care (reference category: in-person visit) in 2022(n = 1930).**

| Socio-demographic and clinical factors | Unadjusted | | Adjusted [a] | |
|---|---|---|---|---|
| | Hybrid care OR (95% CI) | Virtual care OR (95%CI) | Hybrid care OR (95% CI) | Virtual care OR (95%CI) |
| *Age (in years) (ref: > 60)* | | | | |
| ≤ 30 | 0.77 (0.46, 1.29) | 0.48 (0.23, 1.02) | 0.93 (0.26, 3.40) | 0.92 (0.19, 4.43) |
| 31-40 | 1.07 (0.77, 1.48) | 0.87 (0.58, 1.31) | 1.23 (0.47, 3.21) | 1.22 (0.39, 3.76) |
| 41-50 | 1.24 (0.92, 1.66) | 1.24 (0.88, 1.76) | 1.37 (0.69, 2.74) | 1.55 (0.69, 3.46) |
| 51-60 | **1.31 (1.02, 1.69)** | 1.19 (0.88, 1.62) | 1.36 (0.90, 2.06) | 1.20 (0.74, 1.95) |
| *Sex (ref: MSM[1])* | | | | |
| Female | **0.62 (0.48, 0.79)** | **0.48 (0.35, 0.66)** | **0.65 (0.47, 0.90)** | **0.59 (0.40, 0.88)** |
| Heterosexual Men | **0.58 (0.44, 0.76)** | **0.57 (0.41, 0.79)** | **0.54 (0.39, 0.73)** | **0.64 (0.45, 0.92)** |
| *Race/ethnicity (ref: White)* | | | | |
| African/Caribbean/Black | **0.53 (0.42, 0.68)** | 1.00 (1.00, 0.29) | 0.76 (0.55, 1.07) | **0.57 (0.38, 0.87)** |
| Other | **0.62 (0.46, 0.83)** | 0.71 (0.51, 1.00) | **0.67 (0.48, 0.92)** | 0.76 (0.53, 1.11) |
| *Region (ref: Toronto)* | | | | |
| Eastern region | **0.24 (0.15, 0.39)** | **0.44 (0.28, 0.71)** | **0.23 (0.14, 0.37)** | **0.42 (0.26, 0.68)** |
| Northern region | **0.16 (0.04, 0.68)** | 0.66 (0.24, 1.80) | **0.18 (0.04, 0.79)** | 0.74 (0.25, 2.13) |
| Southwestern | **2.48 (1.89, 3.26)** | **1.48 (1.05, 2.11)** | **2.76 (2.01, 3.79)** | **1.67 (1.13, 2.47)** |
| *Language barrier (ref: Fluent in English: Either Canadian born or more than five years of immigration)* | | | | |
| Language barrier | **0.47 (0.34, 0.65)** | **0.26 (0.16, 0.44)** | 0.82 (0.45, 1.50) | 0.73 (0.33, 1.64) |
| *Education level (ref: University education)* | | | | |
| Elementary education | 0.72 (0.37, 1.41) | 0.88 (0.42, 1.89) | 0.74 (0.35, 1.60) | 1.18 (0.52, 2.72) |
| High school | 0.86 (0.66, 1.11) | **0.67 (0.48, 0.94)** | 0.76 (0.56, 1.05) | **0.67 (0.46, 0.98)** |
| College education | 0.97 (0.77, 1.23) | 0.95 (0.72, 1.26) | 0.92 (0.71, 1.19) | 0.95 (0.70, 1.29) |
| *Employment status (ref: Employed)* | | | | |
| Unemployed | 0.96 (0.79, 1.17) | 0.95 (0.75, 1.21) | 0.90 (0.68, 1.19) | 0.89 (0.64, 1.24) |
| *Relationship (ref: Unstable)* | | | | |
| Stable | 0.82 (0.67, 1.01) | 1.01 (0.79, 1.30) | 0.78 (0.62, 1.00) | 0.98 (0.74, 1.30) |
| *Annual gross income (CAD)[2] (ref: ≤ 50,000)* | | | | |
| 51,000-70,000 | 1.02 (0.75, 1.41) | 1.06 (0.73, 1.56) | 0.95 (0.66, 1.35) | 0.87 (0.57, 1.33) |
| 71,000-10,0000 | 0.83 (0.61, 1.13) | 0.77 (0.53, 1.13) | 0.71 (0.50, 1.02) | **0.59 (0.38, 0.91)** |
| > 100,000 | 1.15 (0.90, 1.48) | 1.25 (0.94, 1.68) | 0.99 (0.72, 1.36) | 0.81 (0.55, 1.19) |
| *Years since HIV diagnosis (ref: > 10 years)* | | | | |
| < 5 years | **0.71 (0.52, 0.96)** | **0.31 (0.20, 0.51)** | 0.68 (0.44, 1.03) | **0.36 (0.20, 0.66)** |
| 6–10 years | 0.97 (0.73, 1.30) | 0.69 (0.48, 1.01) | 0.86 (0.61, 1.21) | **0.59 (0.39, 0.91)** |
| *Citizenship and immigration (ref: Born in Canada)* | | | | |
| Immigrant | **0.54 (0.40, 0.73)** | **0.31 (0.22, 0.53)** | 0.72 (0.42, 1.26) | 0.78 (0.38, 1.61) |
| *Privacy available for call (ref: no private space available)* | | | | |
| Privacy available | 0.91 (0.74, 1.13) | 1.17 (0.91, 1.53) | 0.97 (0.75, 1.25) | 1.24 (0.91, 1.68) |
| *Barriers to internet access (ref: no barriers to internet access)* | | | | |
| Internet barriers | 0.65 (0.39, 1.09) | 0.72 (0.39, 1.32) | 0.79 (0.45, 1.41) | 0.93 (0.48, 1.82) |
| *Mental health conditions (ref: people having no mental health conditions)* | | | | |
| Substance use | 1.17 (0.75, 1.80) | 1.04 (0.61, 1.79) | 0.71 (0.43, 1.16) | 0.79 (0.44, 1.42) |
| Mental health condition | **1.24 (1.00, 1.54)** | **1.36 (1.06, 1.76)** | 0.96 (0.71, 1.31) | 0.98 (0.69, 1.40) |
| Depression | 1.12 (0.91, 1.38) | 1.137 (0.89, 1.46) | 1.00 (0.80, 1.26) | 1.03 (0.79, 1.35) |
| General anxiety | 1.11 (0.89, 1.39) | 1.12 (0.86, 1.47) | 1.02 (0.80, 1.31) | 1.07 (0.80, 1.42) |
| PTSD[2] | 1.22 (0.99, 1.50) | 1.08 (0.85, 1.40) | **1.30 (1.03, 1.64)** | 1.17 (0.89, 1.55) |

*(Continued)*

**Table 2.** (Continued)

| Socio-demographic and clinical factors | Unadjusted | | Adjusted [a] | |
| --- | --- | --- | --- | --- |
| | Hybrid care OR (95% CI) | Virtual care OR (95%CI) | Hybrid care OR (95% CI) | Virtual care OR (95%CI) |
| Other | 0.86 (0.70, 1.05) | **0.76 (0.60, 0.97)** | 0.94 (0.72, 1.22) | 0.80 (0.59, 1.09) |
| *Quality of life* | | | | |
| PCS[3] (continuous) | 1.00 (0.99, 1.00) | 1.00 (0.99, 1.01) | 1.00 (0.99, 1.01) | 1.00 (0.99, 1.02) |
| MCS[4] (continuous) | 0.99 (0.98, 1.00) | 0.99 (0.99, 1.01) | 1.01 (1.00, 1.03) | 1.01 (0.99, 1.02) |
| *Alcohol abuse disorder (ref: no harmful alcohol consumption)* | | | | |
| Harmful alcohol | 1.35 (0.95, 1.93) | 1.26 (0.82, 1.95) | 1.04 (0.70, 1.53) | 0.93 (0.58, 1.47) |
| *Adherence to ART[5] (ref: ≥ 95% adherence)* | | | | |
| ART adherence < 95% | **1.25 (1.01, 1.55)** | **1.46 (1.14, 1.88)** | 1.05 (0.83, 1.32) | 1.24 (0.94, 1.62) |
| *Insurance coverage (ref: OHIP)* | | | | |
| No insurance coverage | 0.60 (0.35, 1.05) | **0.16 (0.05, 0.55)** | 1.28 (0.64, 2.58) | 0.48 (0.13, 1.74) |
| *ART coverage (ref: Ontario drug benefit)* | | | | |
| Work | 1.01 (0.79, 1.30) | 1.20 (0.90, 1.62) | 0.89 (0.65, 1.21) | 1.01 (0.71, 1.46) |
| From pocket | 1.20 (0.50, 2.85) | 0.67 (0.19, 2.46) | 1.64 (0.63, 4.31) | 0.88 (0.23, 3.39) |
| Pharmacare | **0.26 (0.08, 0.91)** | 0.33 (0.07, 1.46) | 0.37 (0.10, 1.38) | 0.45 (0.09, 2.14) |
| Other resources | 0.73 (0.48, 1.12) | 0.69 (0.40, 1.18) | 1.11 (0.68, 1.79) | 1.19 (0.65, 2.15) |
| Trillium Drug Program | 0.96 (0.72, 1.27) | 0.98 (0.70, 1.39) | 1.00 (0.72, 1.38) | 0.96 (0.65, 1.41) |
| *Viral load (ref: suppressed/stable VL ≤ 40 copies/mL)* | | | | |
| Unsuppressed viral load | 0.90 (0.66, 1.23) | **0.57 (0.37, 0.89)** | 1.05 (0.74, 1.49) | 0.73 (0.46, 1.15) |
| *Smoker (ref: no or mild smoker)* | | | | |
| Heavy smokers | 0.96 (0.76, 1.21) | 1.14 (0.87, 1.50) | 0.96 (0.75, 1.24) | 1.10 (0.82, 1.46) |
| *Depression (ref: None)* | | | | |
| Mild depression | **1.98 (1.49, 2.64)** | **1.44 (1.02, 2.05)** | **2.20 (1.54, 3.13)** | 1.36 (0.89, 2.09) |
| Moderate depression | **1.60 (1.12, 2.30)** | 1.28 (0.83, 2.00) | **2.06 (1.28, 3.31)** | 1.33 (0.75, 2.35) |
| Moderate -severe depression | **1.78 (1.09, 2.89)** | 1.03 (0.53, 1.99) | **2.80 (1.49, 5.26)** | 1.29 (0.58, 2.89) |
| Severe depression | 1.41 (0.77, 2.61) | 0.68 (0.27, 1.69) | **2.46 (1.13, 5.32)** | 0.80 (0.28, 2.29) |

[a]Adjusted for all variables.

Statistically significant values are bold.

[1]Men who have sex with men.

[2]Canadian Dollars.

[3]Post-traumatic stress disorder.

[4]Physical component summary score.

[5]Mental component summary score.

[6]Antiretroviral therapy.

of virtual care users (n = 220/367) sought care elsewhere (p < 0.05). Most additional appointments involved HIV-related care, including visits to family doctors, HIV specialists, pharmacists, and nurses. Table 3 summarizes the factors influencing virtual care assess.

## Discussion

This study examined three types of care in the 2022 OCS cohort: in-person, virtual, and hybrid care. The findings highlight a significant shift from in-person to virtual care, although in-person visits remained the predominant mode of care in the OCS cohort in 2022. Virtual care was accessed across all demographic and clinical groups, with higher

**Table 3. Factors influencing access to care by the mode of healthcare delivery.**

| Characteristics | In-person care n (%) | Virtual care n (%) | Hybrid care n (%) | Total N (%) | p-value |
|---|---|---|---|---|---|
| Participants | 900 (46.6) | 367 (19.0) | 663 (34.4) | 1930 (100) | |
| *Technology and access* | | | | | |
| Internet access | 840 (93.3) | 335 (91.3) | 640 (96.5) | 1815 (94.0) | 0.106 |
| Private space available | 545 (60.6) | 243 (66.2) | 401 (60.5) | 1189 (61.6) | 0.185 |
| Equipment unaffordable | 6 (0.7) | 3 (0.8) | 1 (0.2) | 10 (0.5) | 0.553 |
| Internet unaffordable | 3 (0.3) | 0 (0.0) | 0 (0.0) | 3 (0.2) | 0.585 |
| Mobile data unavailable | 6 (0.6) | 3 (0.8) | 3 (0.4) | 12 (11) | 0.566 |
| *Barriers to access* | | | | | |
| Appointments interfered with work | 124 (13.8) | 46 (12.5) | 85 (12.8) | 255 (13.2) | 0.783 |
| Virtual care not offered | 355 (40.0) | NA | NA | 355 (40.0) | < 0.001 |
| Preferred In-person care | 624 (69.3) | NA | NA | 624 (69.3) | |
| Travel time too long | 16 (1.8) | 3 (0.3) | 4 (0.6) | 23 (1.1) | 0.081 |
| Public transport unaffordable | 9 (1.0) | 2 (0.2) | 10 (0.9) | 21 (1) | 0.339 |
| No car available | 8 (0.9) | 0 (0.0) | 5 (0.8) | 13 (0.7) | 0.204 |
| Driving costs unaffordable | 24 (4) | 1 (0.1) | 10 (1.5) | 35 (1.8) | 0.011 |
| Other reasons | 3 (2.4) | 0 (0) | 3 (0.4) | 8 (0.4) | 0.204 |
| Total appointments | 3767 | 1395 | 3200 | 8362 | < 0.001 |
| *Healthcare Utilization* | | | | | |
| Total appointments | 3767 | 1395 | 3200 | 8362 | < 0.001 |
| Family doctor visits | 642 (17.0) | 288 (20.6) | 547 (17.0) | 1477 (17.7) | < 0.001 |
| Nurse visits | 436 (11.5) | 105 (7.5) | 291 (9.0) | 832 (9.9) | < 0.001 |
| Pharmacist visits | 378 (10.0) | 124 (8.8) | 311 (9.7) | 813 (9.7) | < 0.001 |
| Other healthcare visits | 860 (22.8) | 404 (28.9) | 1037 (32.4) | 2301 (27.5) | < 0.001 |
| Transferred clinic due to a negative experience | 25 (2.7) | 6 (1.6) | 18 (2.7) | 49 (2.5) | 0.309 |
| Seen other healthcare providers | 600 (39.2) | 220 (54.9) | 454 (41.2) | 1274 (46.4) | 0.05 |
| *Experiences of discrimination* | | | | | |
| Faced discrimination | 73 (8.1) | 31 (8.5) | 58 (8.7) | 162 (8.4) | 0.994 |
| Based on sexual orientation | 8 (1.0) | 3 (1) | 11 (2.0) | 21 (1.0) | NA |
| Based on disability | 0 (0.0) | 0 (0) | 1 (0) | 1 (0.05) | NA |
| Based on HIV status | 49 (5.0) | 19 (5.0) | 36 (5.0) | 104 (5.3) | NA |
| Not treated with dignity | 23 (30.4) | 13 (11.2) | 29 (23.2) | 65 (3.3) | 0.155 |
| Felt stigma | 20 (23.5) | 6 (8.5) | 24 (17.9) | 50 (2.5) | 0.179 |
| Breach of confidentiality | 22 (24.4) | 13 (9.0) | 17 (18.5) | 52 (2.7) | 0.327 |
| Unauthorized info disclosure | 4 (2.4) | 5 (5.0) | 5 (6.6) | 14 (0.7) | 0.479 |

usage observed among residents of Southwestern Ontario. In contrast, in-person care was associated with factors such as being female, heterosexual men, residence in Eastern Ontario, a high school education, and an HIV diagnosis within the past 10 years. Among age groups, individuals aged 51–60 were the most frequent users of virtual care, suggesting greater technological comfort among older adults; however, these differences were not statistically significant (p > 0.097). Patients with depression opted for both virtual and in-person care, likely adjusting their care modality based on their mental health needs. However, the study could not determine whether the choice of care was linked to mental health status.

Prior research has identified several sociodemographic factors influencing the use of different care modalities. Friedman et al. (2022) reported racial and employment-related disparities in telehealth uptake among PLHIV [43], while Baim-Lance et al. (2022) highlighted barriers for older adults related to technology access and digital literacy [44]. A study utilizing Veterans Health Administration Data during the pandemic found that patients with lower income, greater disabilities, or multiple chronic conditions were more likely to use virtual care, whereas those experiencing homelessness were less likely to access it [45].

In contrast, our analysis, conducted after COVID-19 restrictions had eased and virtual care had become more established [46–48], found a stronger preference for in-person care, although patient satisfaction remained high. Several factors associated with choosing in-person visits, including residence in Southern or Eastern Ontario, MSM status, lower (elementary) or higher (university) education levels, and having lived with HIV for more than 10 years, were statistically significant and merit further exploration to better understand patient comfort and inform optimization of HIV care delivery. Notably, 91% of virtual care users reported that their healthcare needs were met, aligning with prior evidence showing that PLHIV receiving both virtual and in-person care expressed high satisfaction when access barriers were minimized [49]. Despite this, barriers remain: 40% of in-person visits occurred because virtual care was not offered, and 70% reflected personal preference.

Virtual care has the potential to advance the UNAIDS 95-95-95 targets by improving access, retention, and adherence to HIV care, particularly for populations facing geographic, socioeconomic, or mobility barriers [50]. Understanding patterns of virtual care use allows healthcare systems to identify underserved groups and implement equity-focused strategies [51]. By complementing in-person services, virtual care supports patient-centered, differentiated care, strengthening the public health response to HIV [49].

### Implications for Practice and Public Health

These findings support the continued use of hybrid HIV care models that integrate both virtual and in-person services. The absence of statistically significant differences in care modality use across age groups and in most access-related barriers suggests that virtual care did not systematically disadvantage older adults or individuals facing technological constraints within this publicly funded healthcare setting. The higher use of hybrid care among individuals with depression highlights the importance of flexible care delivery models that can accommodate fluctuating mental health needs while maintaining continuity of care. From a public health perspective, maintaining multiple modes of care delivery may help promote equitable access, support patient-centred care, and strengthen engagement in HIV services without exacerbating existing disparities.

### Limitations

This study's cross-sectional design limits the ability to draw causal conclusions regarding the relationship between sociodemographic variables and care modality use. Data were also collected during the early stages of virtual care adoption, when standards, infrastructure, and provider training were still developing, which may reduce the applicability of the findings to current practices [46,52].

Although the patient satisfaction survey provides insights into care preferences, it primarily focuses on patient experience and access without capturing provider perspectives, which could introduce bias. Additionally, the study does not examine the usability and functionality of virtual care platforms. Future research should incorporate qualitative interviews and provider surveys to deepen understanding of healthcare professionals' experiences, preferences, and perceived limitations across HIV care modalities.

### Strengths

This study's strengths lie in its comprehensive examination of a broad range of sociodemographic and health-related factors influencing access to care, combined with an in-depth exploration of patient experiences, barriers, and acceptance

of virtual care. By integrating these perspectives, the study provides a detailed overview of virtual care utilization patterns and identifies important areas for future research, as well as actionable opportunities for health system improvements to enhance equitable access and patient-centred care.

Importantly, the robustness of these findings is supported by methodological considerations. Although non-response bias is a common challenge in survey research, our study achieved a 22% response rate (427/1,930), which is sufficient to represent a population of approximately 10,000 with a 5% margin of error [53]. The use of a complete and accurate sampling frame, along with a validated survey instrument, further strengthens the reliability of the results and supports population-level inferences while minimizing random sampling errors [54]. Together, these factors underscore the study's ability to provide meaningful and generalizable insights into virtual care utilization.

Drawing on a large sample and real-time data, the study highlights the importance of long-term research in fully assessing the potential of virtual care. Future research should use longitudinal and administrative data to examine visit patterns, patient and provider preferences, and the cost implications of virtual care to better understand its long-term role in HIV service delivery.

## Conclusions

This study examined the use of virtual care among the 2022 OCS cohort of PLHIV and the socio-demographic factors influencing their care preferences. Nearly half of participants preferred traditional in-person visits, with this preference especially strong among MSM, those with higher education and income, and individuals diagnosed with HIV for more than 10 years. Regardless of the type of care utilized, patients reported high satisfaction with the services received. This exploratory study highlights the potential for further research into different modes of care delivery, emphasizing that effective and equitable integration of technology can strengthen patient-provider relationships, promote fair access, and improve outcomes for PLHIV.

## Supporting information

**S1 Fig. Bar graph on participants' satisfaction with virtual care in the 2022 Ontario HIV Treatment Cohort Study (n = 1,030).**
(TIF)

**S2 Fig. Bar graph on participants' experiences with HIV care providers in the 2022 Ontario HIV Treatment Cohort Study (n = 432).**
(TIF)

**S3 Fig. Bar graph on participants' HIV visits experiences with the clinical practice in the 2022 Ontario Treatment HIV Cohort Study (n = 427).**
(TIF)

**S1 File. STROBE checklist.**
(DOCX)

## Acknowledgments

The OHTN Cohort Study Team consists of Dr. Lawrence Mbuagbaw (Principal Investigator; email: mbuagblc@mcmaster.ca), Department of Health Research Methods, Evidence, and Impact (HEI)

Chair and Principal Investigator; Ontario HIV Treatment Network Cohort Study (OCS); Dr. Anita Benoit (Co-Investigator), University of Toronto; Dr. Sergio Rueda, CAMH and University of Toronto; Dr. Gordon Arbess, Unity Health; Dr. Corinna Quan, Windsor Regional Hospital; Dr. Curtis Cooper, Ottawa General Hospital; Elizabeth Lavoie and Dr.

Maheen Saeed, Byward Family Health Team; Dr. Mona Loutfy and Dr. David Knox, Maple Leaf Medical Clinic; Dr. Nisha Andany, Sunnybrook Health Sciences Centre; Dr. Sharon Walmsley, University Health Network; Dr. Michael Silverman, St. Joseph's Health Care; Tammy Bourque, Health Sciences North; Dr. Marek Smieja, Hamilton Health Sciences Centre; Wangari Tharao, Women's Health in Women's Hands Community Health Centre; Holly Gauvin, Elevate NWO; Dr. Jorge Martinez-Cajas, Kingston Hotel Dieu Hospital; and Dr. Jeffrey Craig, Lakeridge Positive Care Clinic.

We gratefully acknowledge all of the people living with HIV who volunteer to participate in the OHTN Cohort Study. We also acknowledge the work and support of OCS Governance Committee (Aaron Bowerman, Adrian Betts, Barry Adam, Cornel Gray, Dane Record, Jasmine Cotnam, Jason Brophy, Mary Ndung'u, Rodney Rousseau, Ruth Cameron, YY Chen) OCS Scientific Steering Committee (Anita Benoit, Ann Burchell, Barry Adam, Curtis Cooper, David Brennan, Kelly O'Brien, Lance Mcready, Lawrence Mbuagbaw, Mona Loutfy, Pierre Giguere, Sean Hillier, Sergio Rueda (Chair), and Trevor Hart) and Indigenous Data Governance Circle (Meghan Young, Randy Jackson, Trevor Stratton). The OHTN also acknowledges the work of past Governance Committee and Scientific Steering Committee members.

We thank all interviewers, data collectors, research associates, coordinators, nurses, and physicians who provide data collection support. The authors also wish to thank OCS staff for data management, IT support, and study coordination: Lucia Light, Mustafa Karacam, Nahid Qureshi, and Tsegaye Bekele. The Ontario Ministry of Health supports the OHTN Cohort Study.

We also acknowledge the Public Health Laboratories, Public Health Ontario, for supporting record linkage with the HIV viral load database.

The OHTN Cohort Study is supported by the Ontario Ministry of Health and Long-Term Care.

The opinions, results and conclusions are those of the authors and no endorsement by the Ontario HIV Treatment Network or Public Health Ontario is intended or should be inferred.

Realize provided support and guidance for CAB recruitment, consultation on community engagement and study design.

## Author contributions

**Conceptualization:** Nadia Rehman, Lawrence Mbuagbaw, Aaron Jones.

**Data curation:** Nadia Rehman, Lawrence Mbuagbaw, Aaron Jones.

**Formal analysis:** Nadia Rehman.

**Investigation:** Nadia Rehman, Lawrence Mbuagbaw, Dominik Mertz, Giulia M. Muraca, Aaron Jones.

**Methodology:** Nadia Rehman, Lawrence Mbuagbaw, Dominik Mertz, Giulia M. Muraca, Aaron Jones.

**Project administration:** Nadia Rehman, Lawrence Mbuagbaw, Aaron Jones.

**Resources:** Nadia Rehman, Lawrence Mbuagbaw, Aaron Jones.

**Software:** Nadia Rehman, Aaron Jones.

**Supervision:** Lawrence Mbuagbaw, Dominik Mertz, Giulia M. Muraca, Aaron Jones.

**Validation:** Nadia Rehman, Lawrence Mbuagbaw, Aaron Jones.

**Visualization:** Nadia Rehman, Aaron Jones.

**Writing – original draft:** Nadia Rehman.

**Writing – review & editing:** Lawrence Mbuagbaw, Dominik Mertz, Giulia M. Muraca, Aaron Jones.

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
