## [Decision Letter · Decision Letter 0]

21 Oct 2025

PONE-D-25-31000

Socioeconomic determinants of virtual care use among people living with HIV in a clinical cohort in Ontario, Canada: A cross-sectional study

PLOS ONE

Dear Dr. Rehman,

Thank you for submitting your manuscript to PLOS ONE. After careful consideration, we have decided that your manuscript does not meet our criteria for publication and must therefore be rejected.

I am sorry that we cannot be more positive on this occasion, but hope that you appreciate the reasons for this decision.

Kind regards,

Mohammadsobhan S. Andalibi, MD

Academic Editor

PLOS ONE

Reviewer's Responses to Questions

**Comments to the Author**

1. Is the manuscript technically sound, and do the data support the conclusions?

Reviewer #1: Yes

Reviewer #2: Yes

Reviewer #3: Yes

2. Has the statistical analysis been performed appropriately and rigorously?

Reviewer #1: Yes

Reviewer #2: Yes

Reviewer #3: Yes

3. Have the authors made all data underlying the findings in their manuscript fully available?

Reviewer #1: No

Reviewer #2: Yes

Reviewer #3: No

4. Is the manuscript presented in an intelligible fashion and written in standard English?

Reviewer #1: Yes

Reviewer #2: Yes

Reviewer #3: Yes

Reviewer #1: This manuscript provides a detailed overview of Socioeconomic determinants of virtual care use among people living with HIV in a clinical cohort in Ontario, Canada : A cross-sectional study. However, it contains a few errors. The manuscript, in its current form, requires significant improvements. Here are some detailed comments:

1. Is the choice of cross-sectional design appropriate for answering the research question of this study ?

2. Do you consider the socio-economic level measure to be relevant? If so, please explain why.

3. This study sought to understand the factors that influence the types of care modalities used, as this information can help remove barriers to care retention. At the end of this work, can we confirm that the objective was achieved? If so, add a paragraph on the contribution of this work to public health.

4. The study has limitations that may affect the interpretation of the results. In light of this observation, what corrective measures do you envisage to mitigate this major limitation?

Reviewer #2: Thank you for taking time and efforts to study this important topic, kindly consider the following:

Please add a brief description of the burden of HIV in Canada with reference to population SES

Describe the category Men non-MSM

Reviewer #3: PLOS ONE Review

Manuscript ID: PONE-D-25-31000

This manuscript examines the association between socioeconomic factors and the use of virtual care among people living with HIV (PLHIV) in a clinical cohort in Ontario, Canada. The authors assessed the use of virtual care in HIV management among infected patients, thereby providing valuable evidence for future studies and therapeutic interventions aimed at improving treatment adherence and outcomes in this population. The manuscript is generally well-organized, written in clear English, and engaging. The topic is highly relevant to public health practitioners and policymakers, as it provides a framework and rationale for future interventions using digital solutions to improve access to care. However, several sections require clarification and methodological improvements.

Methods Section:

It would be ideal for the authors to follow appropriate research guidelines in the presentation and reporting of their findings. A suitable guideline would be the Strengthening the Reporting of Observational Studies in Epidemiology (STROBE) statement.

Line 106:

The information about the Ontario HIV Treatment Network Cohort Study in reference number 25 is not available in an open-access journal. I recommend briefly describing the cohort directly within the manuscript and citing the reference accordingly.

Results Section:

P-values should be reported in the tables. I suggest using APA style for presenting the findings and including all relevant model parameters for the multinomial logistic regression analysis. Additionally, I recommend reporting p-values for all variables, especially those associated with the adjusted odds ratios.

Discussion Section:

Several cited studies are not properly referenced. Most of the reported findings have not been adequately discussed in the context of whether they support or contradict previous research. Their implications should also be addressed. Please refer to the STROBE guidelines for further guidance.

Acknowledgments Section:

The acknowledgments section is too long. It should be limited to a single paragraph. Please refer to published PLOS ONE articles for formatting examples.

**Do you want your identity to be public for this peer review?** For information about this choice, including consent withdrawal, please see our Privacy Policy

Reviewer #1: No

Reviewer #2: **Yes:** WEAM BANJAR

Reviewer #3: No

- - - - -

---

## [Author Response · Author response to Decision Letter 1]

16 Nov 2025

Subject: Appeal and Revised Submission – Manuscript ID PONE-D-25-31000

Dear PLOS ONE Editorial Team,

Thank you for the opportunity to appeal the rejection decision of our manuscript (PONE-D-25-31000), titled “Socioeconomic determinants of virtual care use among people living with HIV in a clinical cohort in Ontario, Canada: A cross-sectional study,” for reconsideration in PLOS ONE.

We sincerely appreciate PLOS ONE for giving us another opportunity to revise the manuscript, and we are grateful to the reviewers for their time and thoughtful feedback. We have carefully addressed each of their comments and incorporated the suggested changes to enhance the clarity, quality, and scientific rigor of our manuscript.

Below, we provide a detailed, point-by-point response to all comments. The revised manuscript and a marked-up version highlighting the changes are attached. We believe these revisions align with the journal’s standards and hope they meet the expectations of both the reviewers and the editorial team.

Sincerely,

Nadia Rehman

Reviewer’s Comments

Major response: Have the authors made all data underlying the findings in their manuscript fully available?

3. The PLOS Data policy requires authors to make all data underlying the findings described in their manuscript fully available without restriction, with rare exception (please refer to the Data Availability Statement in the manuscript PDF file). The data should be provided as part of the manuscript or its supporting information or deposited to a public repository. For example, in addition to summary statistics, the data points behind means, medians and variance measures should be available. If there are restrictions on publicly sharing data—e.g. participant privacy or use of data from a third party—those must be specified. We thank the reviewers for this comment.

Authors’ Response

We appreciate the importance of PLOS ONE’s data availability policy and have revised the manuscript accordingly.

However, the authors cannot share the dataset directly, as it was obtained under a data-use agreement with the Ontario HIV Treatment Network (OHTN) through the Ontario HIV Treatment Cohort Study (OCS). Access to the data required an extensive review process, including evaluation of the study protocol by eight reviewers, followed by final approval from the OCS Governance Committee. This approval was granted after the protocol was presented at a meeting and after agreement to adhere strictly to the study’s privacy, confidentiality, and dissemination policies. These agreements include strict restrictions that prevent public distribution of the data in any form. However, the data are available to qualified researchers upon request through the OHTN’s Research Application Process (RAP).

We have now included a detailed Data Availability Statement in the manuscript. We previously understood that this information had to be provided only in the application; however, to fully address the reviewer’s concern and ensure transparency, we have now added these details to the manuscript as well.

The following statement has been added to the manuscript:

Data Availability Statement:

“The data used in this study were obtained from the Ontario HIV Treatment Network Cohort Study (OCS). Due to privacy and confidentiality agreements, the authors cannot share the dataset or raw data publicly. However, data access may be requested directly from the OHTN by submitting a Research Application Process (RAP) form. Requests can be made by emailing ocs@ohtn.on.ca.”

Response to Reviewer #1

1.This manuscript provides a detailed overview of Socioeconomic determinants of virtual care use among people living with HIV in a clinical cohort in Ontario, Canada : A cross-sectional study. However, it contains a few errors. The manuscript, in its current form, requires significant improvements. Here are some detailed comments: Is the choice of cross-sectional design appropriate for answering the research question of this study ? (Page 23, Lines 328-332)

Authors response

We thank the reviewer for this comment. This study represents an initial evaluation of the associations between sociodemographic variables and choice of care modality among people living with HIV. Our cross-sectional analysis using multinomial logistic regression identifies statistically significant trends for factors such as gender, region, education and people with depression, indicating that these variables influence whether participants choose virtual or in-person care.

Beyond identifying associations, these findings have important public health and policy implications. Understanding why some individuals prefer one modality over another allows us to assess whether virtual care is equitably accessible to all populations. Equitable access is critical, as progress toward the UNAIDS 95-95-95 targets is hindered by sociodemographic and structural barriers. This work provides foundational insights into potential disparities in care utilization and informs policymakers in developing strategies to improve access and uptake of virtual care among diverse groups of people living with HIV.

While causal inference is not possible with a cross-sectional design, this approach is appropriate for identifying associations and generating hypotheses for future longitudinal, interventional, and qualitative studies, which can further explore how to maximize the benefits of virtual care and reduce barriers to HIV care.

2. Do you consider the socio-economic level measure to be relevant? If so, please explain why.

Authors’ response

We thank the reviewer for this comment. Engagement in the HIV care cascade remains below the UNAIDS 95-95-95 targets in many populations. Virtual care represents one potential strategy to reduce barriers to care, particularly for socioeconomically disadvantaged groups. However, without systematically evaluating its impact and effectiveness, virtual care interventions may inadvertently exacerbate disparities rather than provide benefits. Including socioeconomic measures allows us to assess whether virtual care is equitably accessible and identify populations at risk of being left behind, ensuring that interventions are both effective and inclusive.

3. This study sought to understand the factors that influence the types of care modalities used, as this information can help remove barriers to care retention. At the end of this work, can we confirm that the objective was achieved? If so, add a paragraph on the contribution of this work to public health. We thank the reviewer for this comment.

Authors’ response

Yes, the objective of our study—to understand the factors influencing the choice of care modality among people living with HIV—was achieved. Our analysis identified statistically significant associations between sociodemographic factors, including gender, region, education, and depression, and the use of virtual versus in-person care. These results highlight patterns in care preferences that can inform targeted interventions.

“Virtual care can advance the UNAIDS 95‑95‑95 targets by improving access, retention, and adherence to HIV care, particularly for populations facing geographic, socioeconomic, or mobility barriers [39]. Understanding patterns of virtual care use allows healthcare systems to identify underserved groups and implement equity-focused strategies [40]. By complementing in-person services, virtual care supports patient-centred, differentiated care, strengthening the public health response to HIV [38].” (Page 21, Lines 285-289)

4. The study has limitations that may affect the interpretation of the results. In light of this observation, what corrective measures do you envisage to mitigate this major limitation?

Authors’ response

We thank the reviewer for this observation. We acknowledge that, as a cross-sectional study, causal inference is not possible and that unmeasured confounding or selection bias may influence the observed associations. Additionally, reliance on self-reported data for certain variables could introduce reporting bias.

To mitigate these limitations, we employed adjusted multinomial logistic regression to account for multiple relevant sociodemographic and clinical covariates, thereby reducing confounding. The study employed a large sample size, with real time data. While cross-sectional data cannot establish causality, our study provides an initial evaluation that identifies associations and trends. These findings can guide future longitudinal, interventional, and qualitative studies to more definitively examine causal relationships and further explore barriers and facilitators of virtual care among people living with HIV.

Response to Reviewer#2

1. Please add a brief description of the burden of HIV in Canada with reference to population SES.

Authors response

We thank the reviewer for this comment. We have already provided references describing the Ontario HIV Treatment Cohort Study and the HIV burden in Ontario, which support the representativeness of our findings to the Ontario population. While some aspects of HIV care may differ in other provinces, generalizing these results to all of Canada would not serve the purpose of our study, which is to provide a detailed, context-specific evaluation of virtual care use and sociodemographic determinants in Ontario. These findings remain valuable for informing local public health strategies and can serve as a basis for future studies in other regions. However, there was a major discrepancy that we previously used CDC report 2023 for referencing of retention rates. Now we have made the change and used the Ontario HIV Epidemiology and Surveillance Initiative (OHESI) report, HIV Care Cascade in Ontario: Linkage to Care, In Care, On Antiretroviral Treatment, and Virally Suppressed, 2020. Toronto, ON: OHESI,

“In Ontario, 2020 data show that retention in HIV care varied across regions, with an overall 89.2% of people living with diagnosed HIV retained in care, and lower retention in some areas such as Northwestern Ontario (67.6%).” Page 6-7, lines 101-110

2. Describe the category Men non-MSM

Authors’ response

Thank you for this feedback. We acknowledge that the “Men non-MSM” category was not clearly defined. We have updated the terminology to “Heterosexual Men”, which is universally understood. Additionally, we have changed “Male:MSM” to “MSM” only, as this term is self-explanatory. These changes have been applied consistently in both Table 1 and Table 2, as well as throughout the manuscript. Moreover, we have used the term “hybrid” to replace the “combination of virtual and in-person care”, which is a term used consistently across the literature for people who access both in-person and virtual mediums.

Response to Reviewer #3:

1.This manuscript examines the association between socioeconomic factors and the use of virtual care among people living with HIV (PLHIV) in a clinical cohort in Ontario, Canada. The authors assessed the use of virtual care in HIV management among infected patients, thereby providing valuable evidence for future studies and therapeutic interventions aimed at improving treatment adherence and outcomes in this population. The manuscript is generally well-organized, written in clear English, and engaging. The topic is highly relevant to public health practitioners and policymakers, as it provides a framework and rationale for future interventions using digital solutions to improve access to care. However, several sections require clarification and methodological improvements.

Authors’ response

We sincerely thank the reviewer for their positive and encouraging comments. We appreciate their recognition of the study’s relevance and have addressed the suggested clarifications and methodological improvements in the revised manuscript.

2. Methods section: It would be ideal for the authors to follow appropriate research guidelines in the presentation and reporting of their findings. A suitable guideline would be the Strengthening the Reporting of Observational Studies in Epidemiology (STROBE) statement.

Authors’ response

We thank the reviewer for this helpful suggestion. We had followed the STROBE guidelines during the initial preparation of the manuscript; however, we have now carefully reviewed the checklist to ensure full compliance with all reporting requirements. We have added details on sample size determination and funding sources and have included the completed STROBE checklist as a supplementary file for reference. The changes are given below with specific page numbers.

“Sample size calculations:

As of December 31, 2022, 2155 individuals completed the OCS questionnaire across the ten OCS sites. The primary outcome is adherence to ART, with 692 participants with suboptimal adherence and 1293 participants with optimal adherence. We planned a study with 1930 subjects. The sample size resulted in 80% power to detect a difference of 20% or greater between participants with suboptimal and optimal [35]. The Type I error probability associated with the test of the null hypothesis is 0.05 for two-tailed chi-squared statistic. (PS: Power and Sample Size Calculation version 3.1.2, 2014 by W.D. Dupont & W.D. Plummer Jr).” (Page 11, Lines199-206)

Funding

This study received no funding. (Page 23, Lines 337-338)

3. Line 106: The information about the Ontario HIV Treatment Network Cohort Study in reference number 25 is not available in an open-access journal. I recommend briefly describing the cohort directly within the manuscript and citing the reference accordingly. We thank the reviewer for this helpful feedback. While multiple studies have been published using data from the Ontario HIV Treatment Network Cohort Study (OCS), the most comprehensive description of the cohort is provided in Rourke et al. We have now added a brief summary of the OCS in the manuscript for clarity.

The revised text reads as follows:

“In brief, the source population of the Ontario HIV Treatment Network Cohort Study (OCS) includes people living with HIV aged 16 years and older who receive medical care at specialty clinics within Ontario’s publicly funded healthcare system. Participants are recruited from 15 sites, including hospital-based outpatient clinics and community-based practices. Clinical data from routine follow-up visits are obtained through manual or electronic medical record abstraction, linkage with Public Health Ontario Laboratories (PHOL), the sole provincial provider of HIV-related testing, and annual standardized interviews that capture sociodemographic, psychosocial, and behavioral information. Upon enrolment, participants complete a baseline questionnaire and subsequently an annual follow-up interview addressing demographics, employment, income, mental health, and substance use.” Page 6-7, Lines 102-114

4. Results Section:

P-values should be reported in the tables. I suggest using APA style for presenting the findings and including all relevant model parameters for the multinomial logistic regression analysis. Additionally, I recommend reporting p-values for all variables, especially those associated with the adjusted odds ratios.

Authors’ response

Thank you for your feedback. We have presented the results using both unadjusted and adjusted estimates along with their corresponding confidence intervals and have bolded the significant findings to facilitate easy interpretation.

Due to space constraints, we were unable to provide full p-values or APA-style formatting in the tables. We believe this approach, used for multinomial regression, effectively conveys the key results while remaining concise.

5. Discussion Section:

Several cited studies are not properly referenced.

Most of the reported findings have not been adequately discussed in the context of whether they support or contradict previous research. Their implications should also be addressed.

Please refer to the STROBE guidelines for further guidance.

Authors’ response

We have addressed the reviewer’s comments by enhancing the discussion with relevant and up-to-date references and following the STROBE guidelines to discuss the other evidence, limitations, bias, and applicability of the findings. Page 20-22.

6. Acknowledgments Section:

The acknowledgments section is t

---

## [Decision Letter · Decision Letter 1]

17 Dec 2025

Dear Dr. Rehman,

I have assumed the editorial duties for this manuscript from the previous academic editor. Your manuscript has been re-reviewed by external reviewers who have raised a few minor comments. I invite you to quickly address these points and resubmit. If your responses are satisfactory, I do not anticipate the need for another round of external peer-review. The most important aspect of the comment that I would like for you to focus on is the one regarding the independence assumption. Given that hybrid care is, conceptually, a mix of the other two categories, it cannot necessarily be considered independent from the other two categories and patients would likely not divide evenly should this category be taken away. This could be problematic for your multinomial logistic regression. There are tests that you can perform to assess the validity of this assumption for your analysis, which I invite you to consider completing as a part of your revision. In regards to multicollinearity and nonlinear interactions. I am more interested in the multicollinearity part; given the inclusion of multiple SES variables, the study may benefit from assessment with VIFs. I think given your sample size and the complexity of the work required, an explicit exploration of non-linear interactions is not necessary. Lastly, I agree with the reviewer that a brief comment on public health implications is warranted. Discussions of UNAIDS target and prior studies are good, but suppose I am a clinic manager in Ontario, are there any specific take-aways for me after reading this article?

We look forward to receiving your revised manuscript.

Kind regards,

**Jiawen Deng**

Temerty Faculty of Medicine, University of Toronto

Academic Editor, PLOS One

2. We noted in your submission details that a portion of your manuscript may have been presented or published elsewhere. “We would also like to highlight our recent publication in PLOS ONE, titled “Association between virtual visits and health outcomes of people living with HIV: A cross-sectional study,” based on the 2022 OCS cohort. While both manuscripts draw from the same cohort, they address distinct research questions. Given the differing focuses on aspects of virtual care, the findings are presented in separate publications.” Please clarify whether this [conference proceeding or publication] was peer-reviewed and formally published. If this work was previously peer-reviewed and published, in the cover letter please provide the reason that this work does not constitute dual publication and should be included in the current manuscript.

3. For studies involving third-party data, we encourage authors to share any data specific to their analyses that they can legally distribute. PLOS recognizes, however, that authors may be using third-party data they do not have the rights to share. When third-party data cannot be publicly shared, authors must provide all information necessary for interested researchers to apply to gain access to the data. (https://journals.plos.org/plosone/s/data-availability#loc-acceptable-data-access-restrictions)

4. One of the noted authors is a group or consortium “Ontario HIV Treatment Network Cohort Study”. In addition to naming the author group, please list the individual authors and affiliations within this group in the acknowledgments section of your manuscript. Please also indicate clearly a lead author for this group along with a contact email address.

Reviewers' comments:

Reviewer's Responses to Questions

**Comments to the Author**

Reviewer #1: (No Response)

2. Is the manuscript technically sound, and do the data support the conclusions?

Reviewer #1: Yes

3. Has the statistical analysis been performed appropriately and rigorously?

Reviewer #1: Yes

4. Have the authors made all data underlying the findings in their manuscript fully available?

Reviewer #1: No

5. Is the manuscript presented in an intelligible fashion and written in standard English?

Reviewer #1: Yes

Reviewer #1: This manuscript provides a detailed overview of Socioeconomic determinants of virtual care use among people living with HIV in a clinical cohort in Ontario, Canada : A cross-sectional study. However, it contains a few errors. Consequently, the manuscript, in its current version, requires improvements in certain areas. Here are some detailed comments :

1) We employed multinomial logistic regression to identify

predictors of care mode: virtual, in-person, or a hybrid (virtual and in-person).

How did the authors test the sensitivity to multicolinearity of the independent variables?

How did the authors manage complex nonlinear interactions?

2) We used a three-category multinomial logistic regression, with in-person care as the reference category, to identify independent correlates of virtual care use.

How did the authors manage the limitation related to the hypothesis of independence of irrelevant alternatives?

3) A paragraph on the implications of the results for practice and public health should be added.

**Do you want your identity to be public for this peer review?** For information about this choice, including consent withdrawal, please see our Privacy Policy

Reviewer #1: No

---

## [Author Response · Author response to Decision Letter 2]

31 Dec 2025

Dear Reviewer,

Thank you for the time to review our manuscript and we appreciate the constructive feedback.

Below we have provided a point-by-point response to your comments.

Response to the reviewer’s comments:

4. Have the authors made all data underlying the findings in their manuscript fully available?

The PLOS Data policy requires authors to make all data underlying the findings described in their manuscript fully available without restriction, with rare exceptions (please refer to the Data Availability Statement in the manuscript PDF file). The data should be provided as part of the manuscript or its supporting information, or deposited in a public repository. For example, in addition to summary statistics, the data points underlying mean, median, and variance measures should be available. If there are restrictions on publicly sharing data, e.g. participant privacy or use of data from a third party, those must be specified.

Reviewer #1: No

Authors’ Response:

We obtained the data for this study from the Ontario HIV Treatment Network Cohort Study (OCS). Because the cohort comprises individuals from a vulnerable population, people living with HIV who may also experience social and economic marginalization, access to the data is strictly controlled. The data are provided only following approval of a research protocol by the OCS governance and review committees, contingent upon formal data-sharing and confidentiality agreements. Under these agreements, the authors are not legally permitted to publicly distribute or share the raw dataset.

In accordance with PLOS ONE’s data availability and transparency requirements, we have revised the manuscript to ensure that all required information is clearly provided, enabling other researchers to understand the data source and request access through appropriate channels.

1) A description of the data set and the third-party source.

Authors’ Response:

We have updated the Methods section to include a detailed description of the dataset and its third-party source. Revised text for Page 6-7, Lines 99-116

“We conducted a cross-sectional study using data collected in 2022 from participants enrolled in the OCS. The OCS is an ongoing, multisite clinical cohort of people living with HIV who are receiving HIV care in Ontario, Canada. Detailed descriptions of the cohort have been published previously [22]. Briefly, the source population of the OCS includes individuals aged 16 years and older living with HIV who receive medical care within Ontario’s publicly funded healthcare system. Participants are recruited from 15 HIV care sites across the province, including hospital-based outpatient clinics and community-based practices. The OCS integrates longitudinal clinical data with patient-reported information using standardized data collection procedures.

Clinical data are obtained through manual and electronic abstraction of medical records from routine follow-up visits and are supplemented by linkage with Public Health Ontario Laboratories, the sole provincial provider of HIV-related laboratory testing, including viral load measurements. These clinical data are complemented by standardized questionnaires administered at enrollment and annually thereafter, which collect detailed sociodemographic, socioeconomic, psychosocial, and behavioral information, including employment, income, mental health, substance use, and healthcare utilization [23].

For the present analysis, we used cross-sectional data from the 2022 OCS questionnaire, which included detailed measures of HIV care delivery modalities, including in-person and virtual care via telephone or video, as well as relevant demographic, socioeconomic, and clinical covariates.”

2) If applicable, verification of permission to use the data set:

Authors’ Response:

We have provided documentation verifying permission to use the dataset. Specifically, the confidentiality and data-sharing agreement between the authors and the Ontario HIV Treatment Network has been uploaded and is available under the file titled “OHTN_Confidentiality_Agreement.”

3) Confirmation of whether the authors received any special privileges in accessing the data that other researchers would not have.

Authors’ Response:

The authors did not receive any special privileges in accessing the data. All data were obtained through the standard application and approval procedures established by the Ontario HIV Treatment Network (OHTN). Access to the Ontario HIV Treatment Network Cohort Study (OCS) data is governed by formal data-sharing agreements and ethical approvals, and the same process is available to any qualified researcher who submits a proposal for review and approval by OHTN.

4) All necessary contact information others would need to apply to gain access to the data.

Authors’ Response:

Access to the OCS data is available through a standardized application process. Interested researchers may request access by completing the Research Application Process (RAP) form available on the OHTN website and by contacting the OCS team directly at ocs@ohtn.on.ca. This information is clearly stated in the Data Availability Statement, which reads:

Revised text for Page 24, Lines 378-382

“Data Availability Statement

The data used in this study were obtained from the Ontario HIV Treatment Network Cohort Study (OCS). Due to privacy and confidentiality agreements, the authors cannot share the dataset or raw data publicly. However, data access may be requested directly from the OHTN by submitting a Research Application Process (RAP) form. Requests can be made by emailing ocs@ohtn.on.ca.”

6. Review Comments to the Author

Reviewer #1: This manuscript provides a detailed overview of Socioeconomic determinants of virtual care use among people living with HIV in a clinical cohort in Ontario, Canada: A cross-sectional study. However, it contains a few errors. Consequently, the manuscript, in its current version, requires improvements in certain areas. Here are some detailed comments:

1) We employed multinomial logistic regression to identify

predictors of care mode: virtual, in-person, or a hybrid (virtual and in-person).

How did the authors test the sensitivity to multicollinearity of the independent variables?

How did the authors manage complex nonlinear interactions?

Authors’ Response:

Thank you for this comment. Prior to fitting the multinomial logistic regression models, we conducted a series of diagnostic and preparatory analyses to assess model stability and specification.

Sensitivity to multicollinearity among independent variables was assessed using variance inflation factors (VIFs) derived from a linear proxy model that included the same set of covariates as the multinomial model. All VIF values were below 2, indicating very low collinearity and no evidence of multicollinearity sensitivity in parameter estimates. Corresponding tolerance values were greater than 0.5, further supporting the absence of problematic collinearity.

Regarding complex nonlinear interactions, interaction terms and higher-order nonlinear effects were not included in the final models. Model specification was guided by theoretical and clinical considerations. Based on prior literature and substantive knowledge, key variables, including age, depression severity, and viral load, were modelled as categorical variables to capture potentially nonlinear relationships. The only covariates retained as continuous variables were the Physical Component Summary (PCS-12) and Mental Component Summary (MCS-12) scores, which are validated continuous measures of health-related quality of life.

Plausibility and potential departures from linearity of continuous variables were assessed using graphical methods, including inspection of distributions, residual diagnostics from proxy regression models, and smoothed plots. These assessments did not indicate meaningful departures from linearity or improved model fit with interaction terms. Therefore, a parsimonious main-effects model was retained.

Influential observations were additionally assessed using Cook’s distance; although a small number exhibited elevated influence values, these reflected plausible participant characteristics rather than data errors, and their inclusion did not materially affect model estimates. Relevant clarifications have been added to the Statistical Analysis section. The Model Diagnostics and Specifications read as follows:

“Model Diagnostics and Specification

Prior to fitting the multinomial logistic regression models, diagnostic and preparatory analyses were conducted to assess model stability and specification. Multicollinearity among independent variables was evaluated using variance inflation factors (VIFs) derived from a linear proxy model that included the same set of covariates; all VIFs were below 2, indicating minimal multicollinearity among predictors. Influential observations were assessed using Cook’s distance; although a small number of observations exhibited elevated influence values, these reflected plausible participant characteristics and did not materially affect model estimates.

Theoretical and clinical considerations guided model specification. Variables such as age, depression severity, and viral load were a priori categorized to capture potentially nonlinear relationships supported by the existing literature. The only covariates modelled as continuous variables were the Physical Component Summary (PCS-12) and Mental Component Summary (MCS-12) scores, which are validated continuous measures of health-related quality of life. Plausibility and potential departures from linearity of continuous variables were evaluated using graphical methods, including inspection of distributions, residual diagnostics from proxy regression models, and smoothed plots. These assessments did not reveal meaningful departures from linearity or evidence that inclusion of interaction terms or higher-order nonlinear effects would improve model fit. Accordingly, a parsimonious main-effects multinomial logistic regression model was retained.”

2) We used a three-category multinomial logistic regression, with in-person care as the reference category, to identify independent correlates of virtual care use.

How did the authors manage the limitation related to the hypothesis of independence of irrelevant alternatives?

Authors’ Response:

Thank you for raising this important methodological consideration. We acknowledge the Independence of Irrelevant Alternatives (IIA) assumption inherent to multinomial logistic regression and took several steps to assess the robustness of our findings.

Thank you for raising this important methodological consideration. We acknowledge the Independence of Irrelevant Alternatives (IIA) assumption inherent to multinomial logistic regression and took several steps to assess the robustness of our findings.

Although the Hausman–McFadden test did not indicate statistically significant violations of the Independence of Irrelevant Alternatives (IIA) assumption (p > 0.05), we recognize that this test has known limitations in applied settings, particularly in the presence of sparse cells and complex outcome structures. Therefore, rather than relying solely on this test, we conducted additional sensitivity analyses to evaluate the robustness of our findings.

Prior to model fitting, covariates with limited variability were identified. Housing situation and HIV related stigma showed sparse cell counts across care modality categories. When included in preliminary multinomial models, these variables resulted in numerical instability, including non convergence and undefined coefficient estimates. Consistent with known limitations of multinomial logistic regression under sparse data conditions, these covariates were excluded a priori from the final models to ensure stable estimation and interpretable inference.

Second, we conducted sensitivity analyses by excluding the hybrid care category and re-estimating the models using binary logistic regression comparing virtual versus in-person care. The direction and relative magnitude of associations were largely consistent with those observed in the multinomial models. Older age and privacy concerns were consistently associated with a lower likelihood of virtual care, whereas depression diagnosis, unemployment, higher income, and lower ART adherence were associated with a higher likelihood of virtual care across both modelling approaches. Although effect sizes differed in magnitude, as expected when comparing relative risk ratios and odds ratios, no meaningful discrepancies in interpretation were observed.

Finally, the three outcome categories: in-person care, virtual care, and hybrid care, were defined a priori and represent conceptually distinct modes of healthcare delivery, supporting the plausibility of the IIA assumption in this context. Taken together, these steps support the robustness of the primary findings despite the inherent limitations of the multinomial framework.

Relevant clarifications have been added to the Statistical Analysis section of the manuscript.

Multinomial logistic regression assumes independence of irrelevant alternatives (IIA), in which the relative odds of selecting one outcome category over another are independent of the presence of other alternatives. The revised methods section now reads as follows:

Revised methods section, pages 11-12, Lines 195-224.

“The multinomial logistic regression model relies on the Independence of Irrelevant Alternatives (IIA) assumption. First, we applied the Hausman–McFadden specification test, which indicated no statistically significant violations of the IIA assumption (p > 0.05). However, given the known limitations of this test in finite samples and in models with correlated outcome categories, we additionally conducted a sensitivity analysis excluding the hybrid care category and fitted a binary logistic regression comparing virtual versus in-person care. The direction and magnitude of associations in the sensitivity analysis were consistent with those observed in the primary multinomial model, indicating that the substantive findings were not materially sensitive to potential violations of the IIA assumption. In addition, the outcome categories: in-person care, virtual care, and hybrid care, were defined a priori and represent conceptually distinct modes of healthcare delivery, supporting the appropriateness of the multinomial modelling framework. Covariates with sparse cell counts (housing and stigma) across outcome categories were excluded from the final models, as they led to numerical instability and undefined parameter estimates during multinomial model estimation.”

3) A paragraph on the implications of the results for practice and public health should be added.

Authors’ Response:

Revised discussion on implications of the findings: Page 22-23, lines 332-340

Implications for Practice and Public Health

These findings support the continued use of hybrid HIV care models that integrate both virtual and in-person services. The absence of statistically significant differences in care modality use across age groups and in most access-related barriers suggests that virtual care did not systematically disadvantage older adults or individuals facing technological constraints within this publicly funded healthcare setting. The higher use of hybrid care among individuals with depression highlights the importance of flexible care delivery models that can accommodate fluctuating mental health needs while maintaining continuity of care. From a public health perspective, maintaining multiple modes of care delivery may help promote equitable access, support patient-centred care, and strengthen engagement in HIV services without exacerbating existing disparities.

Authors’ Response:

We have

---

## [Editor Report · Decision Letter 2]

4 Feb 2026

Socioeconomic determinants of virtual care use among people living with HIV in a clinical cohort in Ontario, Canada: A cross-sectional study

PONE-D-25-31000R2

Dear Dr. Rehman,

Your submission is acceptable for publication. Thank you for your revisions. Congratulations!

Please see additional information from the journal attached below.

Regards,

**Jiawen Deng**

Temerty Faculty of Medicine, University of Toronto

Academic Editor, PLOS One

---

## [Editor Report · Acceptance letter]

PONE-D-25-31000R2

PLOS One

Dear Dr. Rehman,

I'm pleased to inform you that your manuscript has been deemed suitable for publication in PLOS One. Congratulations! Your manuscript is now being handed over to our production team.

Kind regards,

on behalf of

Dr. Jiawen Deng

Academic Editor

PLOS One